# Optimizing irradiation dose for *Drosophila melanogaster* males to enhance heterospecific Sterile Insect Technique (h-SIT) against *Drosophila suzukii*

Flavia Cerasti[1]*, Valentina Mastrantonio[1], Alessia Cemmi[2], Ilaria Di Sarcina[2], Massimo Cristofaro[3], Daniele Porretta[1]

1 Department of Environmental Biology, Sapienza University of Rome, Rome, Italy, 2 NUC-IRAD-GAM Laboratory, Italian National Agency for New Technologies, Energy and Sustainable Economic Development (ENEA), Rome, Italy, 3 Biotechnology and Biological Control Agency (BBCA), Rome, Italy

* flavia.cerasti@uniroma1.it

## Abstract

The spotted-wing drosophila (*Drosophila suzukii*), a highly invasive agricultural pest, poses significant challenges to fruit production worldwide. Traditional chemical control methods are costly and raise concerns about resistance and environmental sustainability. The Heterospecific Sterile Insect Technique (h-SIT) has emerged as a promising alternative. Sterile heterospecific males (*Drosophila melanogaster*) can be used to suppress *D. suzukii* populations through reproductive interference, primarily mediated by post-zygotic isolation mechanisms. Although this approach ensures the absence of viable offspring from heterospecific matings, male sterilization through irradiation remains essential. It prevents unintended ecological effects from *D. melanogaster* proliferation in the release area and allows for safe large-scale implementation. Therefore, determining an optimal irradiation dose is critical for achieving high levels of male sterility and maintaining biological quality and mating performance. This study aimed to determine the optimal irradiation dose by assessing induced sterility in *D. melanogaster* males exposed to gamma ray doses ranging from 80–180 Gy. Subsequently, the longevity and the time spent by irradiated *D. melanogaster* males courting *D. suzukii* females were also assessed. Results showed a significant dose-dependent increase in induced sterility, with near-complete sterility at 180 Gy. However, longevity decreased with increasing doses, with males irradiated at 160–180 Gy showing a lifespan reduction of up to 50 days compared to controls. Regardless of the irradiation dose received, *D. melanogaster* males retained their courtship ability toward *D. suzukii* females, although males exposed to 160 Gy exhibited reduced courtship activity. These findings showed that, among the tested doses, 80 Gy was the most effective in preserving male longevity and mating performance, significantly reducing fertility, while 180 Gy induced the highest sterility. The potential

**Data availability statement:** Raw data are now available from Figshare: 10.6084/m9.figshare.28777601.

**Funding:** FC was funded by PhD resources within PON "RICERCA E INNOVAZIONE" 2014–2020", AZIONE IV.5 "DOTTORATI SU TEMATICHE GREEN, D.M. 1061 10 August 2021. DP was funded by Agritech National Research Center and received funding from the European Union Next-Generation EU (PIANO NAZIONALE DI RIPRESA E RESILIENZA (PNRR) – MISSIONE 4 COMPONENTE 2, INVESTIMENTO 1.4 – D.D. 1032 17 June 2022, CN00000022).

**Competing interests:** The authors have declared that no competing interests exist.

lifespan and courtship behavior trade-offs warrant further evaluation. Future studies should evaluate field performance to refine the balance between sterility, longevity, and mating performance for effective *D. suzukii* population suppression.

## 1. Introduction

The spotted-wing drosophila (SWD), *Drosophila suzukii* (Matsumura) (Diptera: Drosophilidae) is an invasive agricultural pest native to Southeast Asia [1]. Since its first detection in California in 2008, *D. suzukii* has rapidly expanded its geographical distribution in many other states of the United States and across the globe, becoming a severe agricultural pest in Europe, South America, and parts of Africa [2–4]. According to recent studies, the economic damage caused by *D. suzukii* in the United States alone reaches hundreds of millions of US dollars annually [5]. In Europe similar losses are reported, with significant economic damage in their fruit industries [6]. The rapid spread of *D. suzukii* has been facilitated by its exceptional ability to thrive in diverse environmental conditions, facilitated by its broad temperature tolerance and adaptability to different habitats [1,7]. One of the factors contributing to the invasive success of *D. suzukii* is its nutritional versatility. *Drosophila suzukii* can attack a broad range of ripening fruits, including soft-skinned berries and stone fruits of economic importance. Adult oviposition and subsequent larval development within fruits lead to rapid fruit decay and increased susceptibility to pathogen infection [8].

To counter the negative impact of *D. suzukii*, efficient and prompt population control interventions are required. Chemical insecticides, especially organophosphates, pyrethroids and spinosyns, have been the most effective control approach, but this strategy is facing multiple challenges [9,10]. Insecticides must be applied several times per growing season, due to *D. suzukii*'s short generation time and larval development inside the fruits [5,9]. Thus, repeated exposure has selected for individuals resistant to insecticides. The short generation time and high fecundity of these resistant individuals have facilitated their rapid population growth, exacerbating the problem and raising concerns about the environmental sustainability of this approach [11,12]. To address this challenge, significant research has been devoted in finding alternative sustainable control measures under an Integrated Pest Management approach [13–16].

In recent years, there has been a renewed interest in the use of SIT (Sterile Insect Technique) and the release of sterile heterospecific males (i.e., heterospecific-Sterile Insect Technique) for pest control [17–20]. Both approaches can be developed under similar theoretical frameworks. Classical SIT consists of releasing large numbers of sterile males of the target pest species into the environment to mate with conspecific wild females. The unfertile mating between the released sterile males and wild females leads to a gradual decline in the pest population over time [21,22]. Contrary to classical SIT, in heterospecific SIT, sterile males from closely related species are released to compete with the pest population for mates. The heterospecific SIT leverages reproductive interference, a reproductive interaction between individuals of

different animal co-generic species and/or subspecies, which results in fitness costs for one or both the interacting individuals [23–27]. It results from incomplete mating barriers between species and can occur at any stage of mate acquisition through different mechanisms, from courtship to mating. These fitness costs typically arise through misdirected mating efforts, gamete wastage, reduced reproductive success, or physical harm (e.g., traumatic insemination) resulting from unsuccessful or incompatible copulation [24,25,27].

A critical aspect shared by both classical and heterospecific SIT approaches is the induction of sterility in males, which is most commonly achieved through irradiation. Consequently, determining the appropriate irradiation dose is essential, as it must induce a high level of sterility while preserving the physiological and reproductive fitness of the treated individuals. Previous studies on various insect pests (e.g., *Ceratitis capitata* Wiedemann, *Anopheles arabiensis* Patton, *Aedes aegypti* L., *Aedes albopictus* Skuse) have demonstrated the importance of conducting dose–response assessments to identify irradiation doses that achieve full sterility without significantly compromising male biological quality [28–32]. At the same time, improperly calibrated irradiation can lead to males that either retain some level of fertility or exhibit impaired mate-finding abilities [29,30], emphasizing the necessity of refining the irradiation dose to balance sterility and biological fitness.

Our previous studies demonstrated that *Drosophila melanogaster* (Meigen) could be a good candidate for *D. suzukii's* control species into a heterospecific SIT context. The two species have incomplete pre-mating and complete post-mating isolation, and reproductive interference has been documented between them under laboratory conditions [19]. Indeed, *D. melanogaster* males successfully courted, mated and inseminated *D. suzukii* females, resulting in reduced fitness of *D. suzukii* [19]. Importantly, although post-zygotic isolation ensures that matings between irradiated *D. melanogaster* males and *D. suzukii* females do not produce viable offspring, male sterilization remains crucial. Irradiation prevents any unintended persistence, reproduction, or ecological interference of *D. melanogaster* itself in the field, enabling safer and more controlled applications of the technique. For this reason, studies were conducted under laboratory conditions to determine the optimal irradiation dose to achieve high levels of male sterility while maintaining biological quality and mating performance. *Drosophila melanogaster* males irradiated at 60 and 80 Gy were able to court and mate with *D. suzukii* females, leading to a significant reduction in *D. suzukii* offspring. However, the induced sterility in *D. melanogaster* males was not complete at these doses [20]. These results provided the first foundation to develop heterospecific SIT against *D. suzukii.*

The aim of this study was to refine and optimize the irradiation dose to enhance the effectiveness of this approach. First, we investigated the effect of 6 different doses from 80 to 180 Gy in inducing sterility of *D. melanogaster* males and assessed their fertility, through mating trials with *D. melanogaster* females. Second, we investigated the effect of the irradiation on male longevity. Finally, we studied the time spent by *D. melanogaster* males irradiated at different irradiation doses, in courting *D. suzukii* females.

## 2. Materials and methods

### 2.1. Fruit fly colonies and rearing techniques

*Drosophila suzukii* and *D. melanogaster* used in this study were routinely reared at the Sapienza University of Rome facilities. The colonies are maintained in the BugDorm-4H4545 insect cages (47.5 x 47.5 x 47.5 cm) in a thermostatic chamber at 25 ± 1°C, with a 14:10 hour light-dark cycle and with a humidity of 70%. The insects were routinely fed with food substrate based on corn flour (84.1% water, 0.7% agar, 3.2% table sugar, 3.6% yeast, 7.2% corn flour, 1.0% soy flour, 0.2% methylparaben dissolved in 25 mL of 70% ethanol) [33]. Each week, the substrate was replaced with a fresh one to allow the insects to feed and lay eggs. The previous substrate was labeled and placed in specified containers to allow the development of new individuals within the colony. The colonies had unrestricted access to water due to cotton balls soaked in a sugar-water solution (1:10 ratio), placed on top of the cages.

## 2.2. *Drosophila melanogaster* males' sterilization and individuals' selection

Sterilization of *D. melanogaster* males was performed at the Calliope gamma irradiation facility at the ENEA Casaccia Research Center (Rome) at different total absorbed doses with a dose rate value of about 130 Gy/h. The Calliope is a pool-type facility equipped with $^{60}$Co radioisotope sources (mean energy 1.25 MeV) in a high volume (7.0 x 6.0 x 3.9 m) shielded cell [34]. To ensure a standardized age for sterilization, males were collected every 30 minutes upon emergence from pupae in breeding falcons. No additional selection was applied beyond emergence timing. In this way, newly emerged males of both species were collected and isolated from females as soon as emergence, ensuring virginity was maintained before experiments. The virgin males collected were placed in separate cages by species. Approximately 72–96 hours after emergence, the virgin adult males were taken for irradiation.

## 2.3. Irradiation effect on *D. melanogaster* male sterility

To evaluate the degree of sterility induced in *D. melanogaster* males at each irradiation dose, mating experiments were conducted between irradiated *D. melanogaster* males and fertile *D. melanogaster* females. Afterward, five sterilized *D. melanogaster* males and five virgin *D. melanogaster* females were placed inside 50 ml falcon tubes containing food substrate. The same procedure was adopted for control individuals. The experiment was conducted in a thermostatic chamber at 25 ± 1°C, with a 14:10 hour light-dark cycle and with a humidity of 70%. After six days, experimental couples were removed, and the emergence of newborns from eggs laid by experimental *D. melanogaster* females mated with *D. melanogaster* males of each condition was monitored. Newly emerged adults were counted and recorded daily. Five replicates were performed for each condition.

## 2.4. Irradiation effect on *D. melanogaster* male longevity

To evaluate the effect of irradiation on the survival of *D. melanogaster* males, the average lifespan between irradiated and non-irradiated *D. melanogaster* males was compared. *D. melanogaster* males aged 72–96 hours old were selected and irradiated at 80 Gy, 100 Gy, 120 Gy, 140 Gy, 160 Gy, 180 Gy, as previously described. An experimental cage (30 × 30 × 30 cm) containing 20 individuals was set up for each dose. Two other cages were set up as controls, in which we placed non-irradiated individuals: one cage was called "home control" with individuals maintained at constant conditions of the thermostatic chamber, and one cage called "trip control", with individuals that we transported to the ENEA Calliope-facility, but outside of the irradiation unit. The "trip control" allowed us to evaluate if the transport could induce an impact on the longevity of the individuals. Mortality was monitored daily under all conditions until all individuals had died.

## 2.5. Irradiation effect on *D. melanogaster* male courtship time

The courtship experiments were conducted to evaluate the time spent courting *D. suzukii* females by *D. suzukii* and irradiated *D. melanogaster* males and to assess potential differences in the courtship time in relation to the administered irradiation doses. The individuals were selected following the methods of the previous analyses. In this experiment, 72–96-hour-old *D. melanogaster* males were irradiated only at 80 Gy, 160 Gy and 180 Gy. Irradiations at 100, 120, and 140 Gy were not considered because they did not lead to significant differences in terms of sterility and longevity (see Results section).

"No-choice" and "choice" experimental trials were set up: in the "no-choice" condition, one *D. suzukii* female was placed with a homospecific or heterospecific male into a falcon (15 mL) and analyzed the male courtship time. Specifically, it was analyzed: – the courtship time of *D. suzukii* male with a *D. suzukii* female; – the courtship time of *D. melanogaster* male irradiated at 80, 160 and 180 Gy with a *D. suzukii* female. In the "choice" conditions, one *D. suzukii* female was placed with two homospecific or heterospecific males into a falcon (15 mL) to evaluate the male's courtship time. Specifically, it was analyzed: – the courtship time of two non-irradiated *D. suzukii* males with a *D. suzukii* female; – the courtship time

of one non-irradiated *D. suzukii* male and one *D. melanogaster* male irradiated at 80, 160 and 180 Gy with a *D. suzukii* female. For the observation of behaviors between two *D. suzukii* males, due to their morphological similarity, the videos were analyzed at reduced playback speed to track the individuals and annotate their behaviors accurately. Conversely, in the second condition involving heterospecific males, the two species were distinguishable: *D. suzukii* males possess characteristic black spots on their wings (hence the name "spotted-wing drosophila"), which are absent in *D. melanogaster* males. For all conditions, following a 5-minute acclimation period, the individual's behavior was recorded for 10 minutes using an Olympus Tough TG-6 camera. After recording, the videos were analyzed using the Boris software (Behavioral Observation Research Interactive Software), taking into account the courtship elements such as orientation, touch, wing scissoring, wing spreading, and copulation attempt [35,36]. A total of 20 replicates were performed for each trial to ensure data robustness.

## 2.6. Data analysis

In the sterility experiment, the sterility degree achieved by *D. melanogaster* males was evaluated for each experimental condition, applying a GLM model (generalized linear model with negative binomial distribution), and the model family was selected comparing the AIC and BIC estimators and the likelihood ratio test. We performed Tukey's multiple comparison test as a post hoc test using the '*multcomp*' package [37]. The average percentage of residual fertility in each condition was obtained by calculating the percentage reduction of each replicate in a specific condition compared to the mean of offspring born in the control condition (i.e., 100% fertility) and calculating the mean (± SE) of the percentages obtained in each replicate.

To evaluate the effect of the irradiation dose on the longevity of *D. melanogaster* males, survival distributions of the different *D. melanogaster* groups ('Control home, 'Control trip', '80 Gy', '100 Gy', '120 Gy', '140 Gy', '160 Gy', '180 Gy') were computed using the Kaplan-Meier method with the 'survival' package and the differences between survival distributions were estimated using the Log-Rank Test with the '*survminer*' package [38,39].

In the courtship experiment, to compare the courtship time of *D. suzukii* and *D. melanogaster* males in "no-choice" condition, we used a GLM model (generalized linear model with negative binomial distribution), selecting the model family based on the AIC and BIC estimators and the likelihood ratio test. We performed Tukey's multiple comparison test as a post hoc test. In the "choice" condition, we used a GLM model (generalized linear model with negative binomial distribution) to compare the average courtship time of males. Then, we compared the average courtship time of the two males in the same condition using the nonparametric statistical Wilcoxon Signed Rank test using the '*dplyr*' package [40]. All analyses were carried out using R Software version 3.6.2. [41].

## 3. Results

### 3.1. *Drosophila melanogaster* males' sterilization

We assessed the induced sterility of *D. melanogaster* males by mating them with *D. melanogaster* females at irradiation doses of 80, 100, 120, 140, 160, and 180 Gy, with a non-irradiated control group for comparison. The mean number of the emerged adults from experimental food substrates in the control condition was 171.8 (± 24.47) (mean ± SE) (Table 1). A significant emergence reduction was observed at all irradiation doses. At the 80 Gy irradiation condition, the mean number of emerged adults was 29.2 (± 9.25), while at 180 Gy irradiation, it dropped to 0.8 (± 0.58) (Table 1; Fig 1). The GLM model showed a significant effect of the male irradiation dose on the number of offspring produced by *D. melanogaster* females (Table 2). Tukey's multiple comparison test showed a significant offspring reduction from the control condition to all irradiation conditions, i.e., 80 Gy (z = 4.253, p < 0.001), 100 Gy (z = 5.183, p < 0.001), 120 Gy (z = 5.634, p < 0.001), 140 Gy (z = 6.660, p < 0.001), 160 Gy (z = 7.194, p < 0.001) and 180 Gy (z = 8.317, p < 0.001). There were also significant differences between the 80 Gy irradiation dose and both the 160 Gy (z = 3.497, p = 0.008) and 180 Gy (z = 5.535, p < 0.001)

**Table 1. The mean number of the emerged *D. melanogaster* adults.** Mean number (±SE) of emerged adults and average percentage (±SE) of the residual fertility at the different treatment doses (Gy).

| Treatment Dose (Gy) | Mean number of emerged adults (±SE) | Average percentage (±SE) of residual fertility |
|---|---|---|
| 0 Gy | 171.8 (± 24.47) | 100% |
| 80 Gy | 29.2 (± 9.25) | 17% (± 5.38) |
| 100 Gy | 23.5 (± 5.52) | 11.29% (± 3.45) |
| 120 Gy | 19.5 (± 2.25) | 9.19% (± 2.38) |
| 140 Gy | 11.75 (± 2.66) | 5.59% (± 1.73) |
| 160 Gy | 5.25 (± 2.39) | 3.05% (± 24.47) |
| 180 Gy | 0.8 (± 0.58) | 0.47% (± 0.34) |

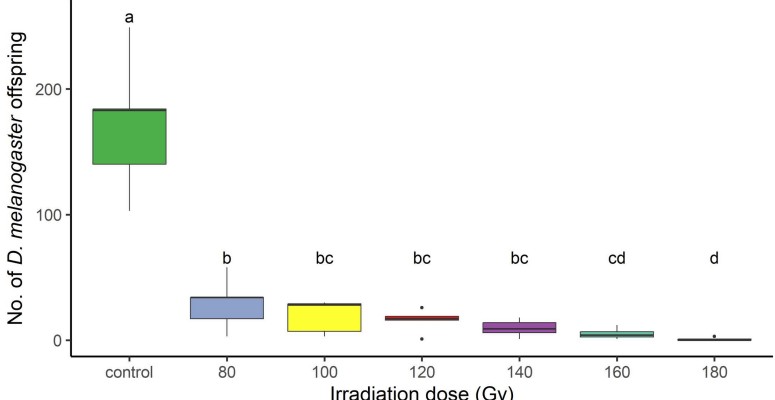

**Fig 1. Irradiation effect on *D. melanogaster* male sterility.** The number of *D. melanogaster* adults emerged from fertile females and irradiated males. Different letters mean significant differences by Tukey Multiple Comparison tests (p<0.05).

doses. The 180 Gy dose showed significant differences with 80 Gy (see above), 100 Gy (z=−4.886, p<0.001), 120 Gy (z=−4.559, p<0.001) and 140 Gy (z=−3.761, p=0.003) doses, but not with 160 Gy. We did not observe significant differences among 80, 100, 120 and 140 Gy, nor among 100, 120, 140 and 160 Gy (p>0.05) (Fig 1).

### 3.2. Irradiation effects on *D. melanogaster* male longevity

In the longevity tests of *D. melanogaster* males, Kaplan-Meier curves showed significant differences in lifespan between the treatments (80 Gy, 100 Gy, 120 Gy, 140 Gy, 160 Gy, 180 Gy, the 'home control' and 'trip control' conditions) (Mantel-Cox log-rank; $\chi2=105.9$, d.f.=7, p<2e-16) (Fig 2). The pairwise comparison test showed that control individuals have a higher probability of survival than irradiated individuals. In particular, the 'trip control' condition showed significant differences with all the conditions tested, while the 'home control' condition showed significant differences only with 160, 180 Gy and the 'trip control' condition (p<0.05). Comparisons between irradiated groups showed that survival generally decreased with increasing dose, with significant differences observed especially between the doses from 80 to 140 Gy and the two highest doses (160–180 Gy) (p<0.05). No significant differences in survival were detected from 80 to 140 Gy doses (p>0.05). Significant differences were also observed between the highest doses tested (160 and 180 Gy) (p<0.05) (Table 3). The average lifespan of individuals in the trip and control conditions was 71 and 68 days, respectively. The average life of males irradiated at 80, 100, 120 and 140 Gy was 63 days, while the males irradiated at 160 and 180 Gy had an average life of 46 and 55 days, respectively (Fig 2).

**Table 2.  Irradiation effect on *D. melanogaster* male sterility. GLM model values are shown. Values in boldface indicate significant differences (p < 0.05).**

| Fixed Effects | Estimate | ±SE | z Value | *p*-Value |
|---|---|---|---|---|
| (Intercept) | 2.9653 | 0.3051 | 9,718 | **<2e-16** |
| 120 Gy | −0.2053 | 0.4342 | −0.473 | 0.63643 |
| 140 Gy | −0.7035 | 0.4435 | −1.586 | 0.11272 |
| 160 Gy | −1.3070 | 0.4942 | −2.645 | **0.00817** |
| 180 Gy | −3.1884 | 0.6526 | −4.886 | **1.03e-06** |
| 80 Gy | 0.4089 | 0.4275 | 0.956 | 0.33882 |
| No irradiation | 2.1811 | 0.4208 | 5.183 | **2.18e-07** |

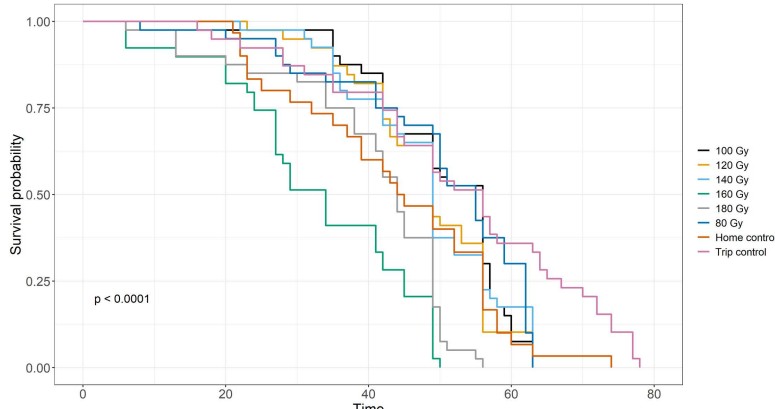

**Fig 2.  Irradiation effect on *D. melanogaster* male longevity.** Kaplan-Meier curves showing the effect of different irradiation doses on the longevity of *D. melanogaster* males.

### 3.3.  Irradiation effect on *D. melanogaster* male courtship time

In the "no-choice" conditions, the mean (± SE) courtship time widely ranged from 16.70% (± 3.98) (*D. melanogaster* irradiated at 160 Gy) up to 67.66% (± 8.30) (*D. suzukii* males) (Fig 3). The GLM model showed significant differences in the average courtship time among different irradiation treatments (Table 4). Tukey's multiple comparison tests showed a significant difference between *D. suzukii* homospecific condition and the conditions with *D. melanogaster* irradiated at 160 Gy (z = 4.346, p < 0.001) and 180 Gy (z = 3.552, p = 0.002) and also a significant difference between *D. melanogaster* irradiated at 80 Gy and 160 Gy (z = 2.697, p = 0.035). No significant differences were observed in courtship time between *D. suzukii* males and *D. melanogaster* males irradiated at 80 Gy, nor between *D. melanogaster* irradiated at 80 and 180 Gy, nor between those irradiated at 160 and 180 Gy (p > 0.05) (Fig 3).

In the "choice" trials, the mean courtship times varied across treatments. In the conspecific condition with two *D. suzukii* males, the mean courtship time was 24.23% (± 6.17) for one male and 16% (± 3.50) for the other. In the heterospecific condition with irradiated *D. melanogaster* males at 80 Gy, the *D. suzukii* male exhibited a mean courtship time of 12.60% (± 3.06), while the *D. melanogaster* male displayed 31.83% (± 7.37). In the heterospecific condition with irradiated *D. melanogaster* males at 160 Gy, the *D. suzukii* male showed a mean courtship time of 24% (± 4.88), whereas the *D. melanogaster* male had 9.66% (± 4.61). In the heterospecific condition with *D. melanogaster* males irradiated at 180 Gy, the courtship time was 12.59% (± 1.94) for the *D. suzukii* male and 13.58% (± 5.68) for the *D. melanogaster* male (Fig 4).

**Table 3. Pairwise comparison of adult longevity across treatments.** P-values from pairwise comparisons between control conditions and irradiation doses. Values in boldface indicate significant differences (p<0.05).

| | Trip control | 80 Gy | 100 Gy | 120 Gy | 140 Gy | 160 Gy | 180 Gy |
|---|---|---|---|---|---|---|---|
| **Home control** | **0.014** | 0.092 | 0.271 | 0.563 | 0.383 | **0.001** | **0.027** |
| **Trip control** | | **0.049** | **0.027** | **0.013** | **0.017** | **<0.001** | **<0.001** |
| **80 Gy** | — | | 0.400 | 0.318 | 0.601 | **<0.001** | **<0.001** |
| **100 Gy** | — | — | | 0.405 | 0.861 | **<0.001** | **<0.001** |
| **120 Gy** | — | — | — | | 0.780 | **<0.001** | **<0.001** |
| **140 Gy** | — | — | — | — | | **<0.001** | **<0.001** |
| **160 Gy** | — | — | — | — | — | | **0.013** |

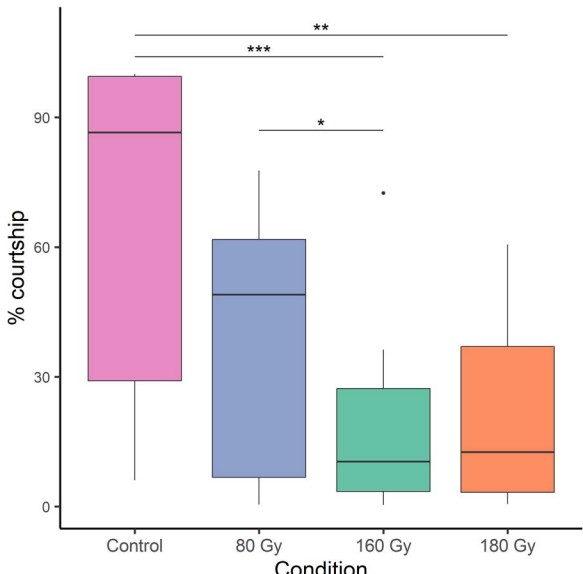

**Fig 3. Courtship comparison in the "no-choice" trials.** Courtship of *D. suzukii* males toward *D. suzukii* females (pink column); courtship of *D. melanogaster* males toward *D. suzukii* females at different irradiation doses (blue, green and orange columns). *** Tukey's multiple com-parison tests p<0.001; ** Tukey's multiple comparison tests p<0.01; * Tukey's multiple comparison tests p-value<0.05. Black dots are box-plot outliers.

The GLM analysis revealed significant differences in the average courtship time among conditions (Table 5), and Tukey's multiple comparisons test indicated a significant difference only between the courtship times of *D. melanogaster* males irradiated at 80 Gy and 160 Gy (p=0.016; Fig 4). The Wilcoxon rank sum test revealed significant differences in courtship time only between *D. melanogaster* males irradiated at 160 Gy and *D. suzukii* males (W=82.5, p=0.002) (Fig 4).

## 4. Discussion

Finding the best irradiation dose is a crucial step that requires careful evaluation to develop a heterospecific SIT approach. We found that irradiation was highly effective at inducing sterility. All irradiation doses led to a significant reduc-tion in adult emergence with respect to the control condition (Table 1; Fig 1). We observed at the lower irradiation dose tested (80 Gy) only a 17% (± 5.38) average residual fertility that decreases as the irradiation doses increase until reaching 0.47 (± 0.34) average residual fertility at the highest dose tested (180 Gy) (Table 1). These results are consistent with previous findings. Studies about the effect of gamma rays on the sterility of *D. melanogaster* were carried out in the 1960s

**Table 4. Irradiation effect on *D. melanogaster* male courtship time in "no-choice" condition. GLM model values are shown.** Values in boldface indicate significant differences. *D. suz = Drosophila suzukii*; *D. mel = Drosophila melanogaster*.

| Fixed Effects | Estimate | ±SE | z Value | *p*-Value |
|---|---|---|---|---|
| (Intercept) | 2.8151 | 0.2273 | 12,384 | **<2e-16** |
| *D. mel 180 Gy* | 0.2592 | 0.3204 | 0.809 | 0.41853 |
| *D. mel 80 Gy* | 0.8595 | 0.3188 | 2.697 | **0.00069** |
| *D. suz* (homospecific condition) | 1.3994 | 0.3220 | 4.346 | **1.39e-05** |

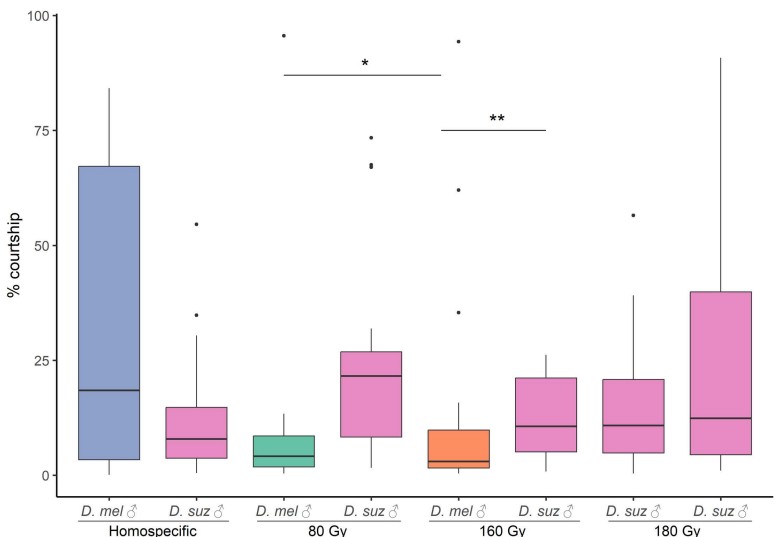

**Fig 4. Courtship comparisons in "choice" trials.** Time spent courting *D. suzukii* females by *D. suzukii* males and irradiated *D. melanogaster* males at 80, 160 and 180 Gy. *D. suz = D. suzukii* males (pink columns); *D. mel 80 Gy = D. melanogaster* males irradiated at 80 grey (blue column); *D. mel 160 Gy = D. melanogaster* males irradiated at 160 grey (green column); *D. mel 180 Gy = D. melanogaster* males irradiated at 180 grey (orange column). Black dots are box-plot outliers. * Tukey's multiple comparison tests p-value < 0.05; ** Wilcoxon rank sum test p < 0.01.

and '70s. Henneberry (1963) [42] found that *D. melanogaster* irradiated males at 160 Gy dose produced nonviable eggs after mating with non-irradiated females. Accordingly, Nelson (1973) [43] observed that at 120 Gy 99.3% fewer progeny emerged than from non-irradiated individuals.

A trade-off between the sterility and longevity of the irradiated males is critical in optimizing classic and heterospecific SIT applications [44]. The survival analysis showed that irradiation significantly reduces the lifespan of *D. melanogaster* males, with the highest reductions in longevity at the highest doses. Control individuals lived on average for 70 days, whereas males irradiated at the highest doses (160–180 Gy) experienced a 50-day lifespan (Fig 2). This observation aligns with prior studies indicating that irradiation-induced oxidative stress and cellular damage can impair physiological functions, shortening lifespan [45]. Nelson et al. (1973) [43] also reported decreased longevity in irradiated *D. melanogaster*, with a similar reduction in lifespan at the highest dose tested of 150 Gy. The dose-dependent decrease in longevity must be carefully considered when applying SIT since male competitiveness may be compromised if they do not survive long enough to mate effectively in the wild. The average lifespan of *D. melanogaster* decreased from 70 days in the control group to 50 days at the highest radiation doses, a reduction that is unlikely to compromise the effectiveness of the SIT approach, as frequent releases of sterile individuals are typically part of the strategy. For instance, regarding screwworm

**Table 5. Irradiation effect on *D. melanogaster* male courtship time in "choice" condition. GLM model values are shown. Values in boldface indicate significant differences. *D. suz* = *Drosophila suzukii*; *D. mel* = *Drosophila melanogaster*.**

| Fixed Effects | Estimate | ±SE | z Value | p-Value |
|---|---|---|---|---|
| (Intercept) | 3.4604 | 0.2447 | 14,139 | **<2e-16** |
| *D. suz* 80 Gy | −0.9271 | 0.3496 | −2.652 | **0.00799** |
| *D. mel* 160 Gy | −1.1924 | 0.3513 | −3.394 | **0.00068** |
| *D. suz* 160 Gy | −0.2828 | 0.3469 | −0.815 | 0.41493 |
| *D. mel* 180 Gy | −0.8515 | 0.3538 | −2.407 | **0.01609** |
| *D. suz* 180 Gy | −0.9279 | 0.3496 | −2.654 | **0.00794** |
| *D. suz* 1 (homospecific condition) | −0.6934 | 0.3530 | −1.964 | **0.04948** |
| *D. suz* 2 (homospecific condition) | −0.2730 | 0.3514 | −0.777 | 0.43724 |

*Cochliomyia hominivorax* (Coquerel), the releases have to occur weekly to maintain the critical ratio or even twice a week for the Mediterranean fruit fly and tsetse *Glossina austeni* (Wiedemann) [46–48]. It is important to note that this study was not addressed to assess the longevity of sterile individuals under field conditions, which can be lower than in protected field-cage situations, where sterile males have easy access to food and are protected from predation in the laboratory [49]. This aspect certainly warrants further investigation.

The last part of this study was designed to assess the heterospecific courtship behavior of irradiated *D. melanogaster* males. A balance between sterility and behavioral competence when selecting an irradiation dose for pest control is critical. If males lose the ability to court females, their sterility will have limited impact on population suppression, as seen in studies of other insect species, such as the Queensland fruit fly (*Bactrocera tryoni* Froggatt), where high doses impaired sexual competitiveness [50]. Our findings showed that irradiated *D. melanogaster* males retained their ability to court *D. suzukii* females even at the highest irradiation doses, suggesting that courtship behavior remains largely unaffected by irradiation. In the "no-choice" condition, however, *D. melanogaster* males irradiated at 160 and 180 Gy exhibited significantly lower courtship activity compared to *D. suzukii* males toward conspecific females (Table 4; Fig 3). Conversely, under the "choice" condition, *D. melanogaster* males courted *D. suzukii* females as much as *D. suzukii* males, even at the highest irradiation doses tested. Notably, *D. melanogaster* males irradiated at 160 Gy showed reduced courtship toward *D. suzukii* females compared to *D. melanogaster* males irradiated at 80 Gy and *D. suzukii* males, corroborating the observations made in the "no-choice" condition (Fig 4). These results suggest two key points. First, the presence of *D. melanogaster* males seems to influence the courtship behavior of *D. suzukii* males, as they courted conspecific females more in the "no-choice" condition compared to the "choice" condition. The reduced courtship behavior observed at a radiation dose of 160 Gy suggests that higher doses may lead to behavioral impairments in *D. melanogaster* males. These impairments are likely attributable to physiological alterations or disruptions in neural circuits essential for mating displays. Ionizing radiation is known to damage neural pathways involved in courtship behavior, as evidenced in moths, where higher doses often result in physiological defects that reduce their competitiveness with wild populations [51,52]. Radiation may also interfere with producing or expressing key biochemical and behavioral signals. During courtship, *D. melanogaster* males emit specific biochemical signals, such as cis-vaccenyl acetate, along with behavioral signals like wing vibrations and pheromone release, to stimulate female responses [53–55]. Ionizing radiation may disrupt these signals, compromising the male's ability to communicate with females effectively. Similar disruptions have been observed in other pest species, such as *Callosobruchus chinensis* L. females and *Anthonomus grandis* (Boheman) males, where radiation-induced impairments in mating signals led to reduced courtship and mating success [56,57]. Consequently, irradiated *D. melanogaster* males may fail to elicit appropriate responses from females, disrupting courtship and mating dynamics. In our

study, while *D. melanogaster* males irradiated at 160 Gy showed reduced courtship, males exposed to 180 Gy courted *D. suzukii* females comparably to untreated *D. suzukii* males under both "no-choice" and "choice" conditions (Figs 3 and 4). Additionally, the highest courtship percentage was observed at the lowest tested dose of 80 Gy (Figs 3 and 4), supporting the notion that lower radiation doses may preserve male courtship behavior more effectively.

Overall, our study highlights the complex interactions between irradiation, longevity, sterility, and mating behavior in *D. melanogaster*, contributing to the growing evidence of using heterospecific SIT in pest control. Based on our results, the 80 Gy and 180 Gy radiation doses appear most suitable for further investigation. At 80 Gy, we observed the highest courtship rates and longest lifespan among the radiation doses tested. In comparison, at 180 Gy, we achieved the greatest reduction in fertility.

For an effective SIT program, it is essential that sterile males survive for a long time in their environment. Only in this way they can mate with a sufficient number of wild females and induce sterility in the population. If their quality is compromised and their longevity reduced, more frequent and larger-scale releases will be required to sustain a high overflooding ratio, ultimately increasing operational costs [58]. Studies on *An. arabiensis* and *Ae. aegypti* have highlighted different factors influencing longevity. *An. arabiensis* was found to have a significantly shorter lifespan in field settings compared to laboratory conditions, whereas *Ae. aegypti* exhibited a stronger sensitivity to seasonality [59]. Specifically, in a field experiment carried out in Vietnam, *Ae. aegypti* populations demonstrated significantly higher survival rates during the cool or hot dry seasons compared to the cool and wet seasons [60]. Moreover, a synergistic effect of irradiation, packing, and chilling was observed to compromise the longevity of *An. arabiensis* —an effect that was not detected in *Ae. aegypti* [59]. These results suggest that the impact of irradiation on lifespan is highly species-dependent. In our case, it is essential first to assess the differences in *D. melanogaster* males' longevity kept under laboratory conditions and those exposed to environmental conditions. This would allow us to better evaluate the lifespan reduction at 180 Gy and determine whether it constitutes a limitation for SIT applications. Conversely, a higher residual fertility at 80 Gy than 180 Gy might be less restrictive given that we are not dealing with a pest or invasive species. Unlike in standard SIT applications for pest control, achieving the high sterility levels required to release pest males (as discussed by Bakri et al. 2021 [61] and Parker et al. 2021 [62]) may not be necessary in this context. Based on these considerations, the present study has demonstrated that an irradiation dose of 80 Gy seems to be more effective. However, further studies are needed better to evaluate both longevity and mating performance under field conditions.

Greenhouses and other enclosed environments appear ideal for implementing the heterospecific Sterile Insect Technique (h-SIT) to manage *D. suzukii* populations. Studies on plastic- and mesh-covered tunnels have shown that mechanical barriers alone can significantly reduce *D. suzukii* populations in these confined areas. This reduction is due not only to the physical exclusion provided by the barriers but also to creating an unfavorable microclimate for the pest's survival [63]. Although complete exclusion cannot be achieved through mechanical barriers alone, integrating h-SIT with these measures could enhance the overall effectiveness of biocontrol strategies.

## 5. Conclusions

Our findings highlight the critical balance between sterility, longevity, and mating behavior in *D. melanogaster* for heterospecific SIT applications. Among the tested doses, 80 Gy emerged as the most effective, preserving male longevity and mating performance while significantly reducing fertility. While 180 Gy achieved the highest sterility, the potential lifespan and courtship behavior trade-offs warrant further evaluation. Future studies should focus on-field performance to refine SIT protocols. Integrating h-SIT with mechanical barriers in controlled environments like greenhouses could enhance *D. suzukii* management, making 80 Gy a promising dose for practical implementation.

## Acknowledgments

We thank Alessandra Spanò, Elisa Michelangeli and Giulia Pezzi for their technical help.

## Author contributions

**Conceptualization:** Flavia Cerasti, Massimo Cristofaro, Daniele Porretta.

**Data curation:** Flavia Cerasti.

**Formal analysis:** Flavia Cerasti, Valentina Mastrantonio.

**Investigation:** Flavia Cerasti, Valentina Mastrantonio.

**Methodology:** Alessia Cemmi, Ilaria Di Sarcina.

**Supervision:** Daniele Porretta.

**Validation:** Flavia Cerasti.

**Visualization:** Flavia Cerasti.

**Writing – original draft:** Flavia Cerasti.

**Writing – review & editing:** Flavia Cerasti, Valentina Mastrantonio, Alessia Cemmi, Ilaria Di Sarcina, Massimo Cristofaro, Daniele Porretta.

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
