## [Decision Letter · Decision Letter 0]

Dear Dr. CERASTI,

Thank you for submitting your manuscript to PLOS ONE. After careful consideration, we feel that it has merit but does not fully meet PLOS ONE’s publication criteria as it currently stands. Therefore, we invite you to submit a revised version of the manuscript that addresses the points raised during the review process.

We look forward to receiving your revised manuscript.

Kind regards,

Herman Wijnen, Ph.D.

Academic Editor

PLOS ONE

Journal Requirements:

2. In the online submission form, you indicated that raw data will be available from the authors under request.

Additional Editor Comments: 

Please respond to each of the reviewers' stated comments and suggestions. Where arguments or language were insufficiently clear to the reviewer(s), I would expect to see revisions. There is no obligation to follow other stylistic suggestions.

In addition:

1) The quality of the images in the Figures needs to be improved (also pick less closely related colors for the lines Fig 2).

2) Reference 20 has a different title than the apparent BIORXIV preprint match 'Developing heterospecific Sterile Insect Technique for pest control: insights from the spotted wing fly Drosophila suzukii'. Please explain. What is the status of this manuscript as well as its relationship to this submission? Please confirm that there is no overlap in the data.

3) You will need to fix your data availability statement. Making raw data only available upon request is not compatible with your answer 'Yes' to the data availability question. You can use supplements or a public data repository to fix this or answer 'No' to the question and provide acceptable justification.

Reviewers' comments:

Reviewer's Responses to Questions

**Comments to the Author**

1. Is the manuscript technically sound, and do the data support the conclusions?

Reviewer #1: Yes

Reviewer #2: Yes

2. Has the statistical analysis been performed appropriately and rigorously?

Reviewer #1: Yes

Reviewer #2: Yes

3. Have the authors made all data underlying the findings in their manuscript fully available?

Reviewer #1: Yes

Reviewer #2: Yes

4. Is the manuscript presented in an intelligible fashion and written in standard English?

Reviewer #1: Yes

Reviewer #2: No

Reviewer #1: The study focuses on optimizing the irradiation dose for Drosophila melanogaster males to achieve maximum sterility while minimizing the impact on insect fitness. The authors attempt to correlate these findings with their previous research on the Heterospecific Sterile Insect Technique (hSIT) against Drosophila suzukii. However, no clear relationship between the current study and their previous hSIT findings was established.

If an integrated approach combining hSIT and irradiation is required for managing D. suzukii, the experiments should optimize irradiation doses following heterospecific mating. This would allow for a comprehensive evaluation of key parameters, including fertility, sterility, fitness, and courtship behaviors. If fitness is compromised in D. melanogaster males during heterospecific mating (line 68-72) with D. suzukii females, the males are already weakened before undergoing irradiation. Thus, for a more accurate assessment of cumulative fitness trade-offs, D. melanogaster males should undergo irradiation after heterospecific mating rather than using untreated males.

In my view, the current findings cannot be directly correlated with previous hSIT experiments. Below are my main suggestions to enhance the study's impact:

1. Title Revision: The title should be rephrased for clarity. I suggest the following: "Optimizing Irradiation Dose for Drosophila melanogaster Males to Enhance Heterospecific Sterile Insect Technique Against Drosophila suzukii."

2. Clarify the Study's Importance: If hSIT alone results in complete sterility, why is an optimized irradiation dose necessary? Please address the significance of this study in both the abstract and introduction sections.

3. Expand on Fitness Costs (Introduction, Line 68): The authors mention that hSIT results in fitness costs for one or both interacting individuals. Please elaborate on how these fitness costs manifest and identify which traits are affected. If flight ability is compromised, it could reduce the feasibility of field releases.

4. Quantify Offspring Reduction (Line 87): Please specify the level of offspring reduction achieved in your previous hSIT studies against D. suzukii. If 100% sterility was not achieved through hSIT alone, highlight how this justifies the need for irradiation to enhance hSIT's effectiveness.

5. Clarify Behavioral Analysis: Since courtship behavior is complex, and you specifically analyze the time males spend with females, it would be more precise to state: "We analyzed the time spent by male D. melanogaster with D. suzukii females" rather than using the broader term "courtship behavior.

Reviewer #2: The science and experimental design is appropriate for the research questions being addressed. However, there is a need to improve on the presentation of information including methods, results and discussion. The major concern is on the writing style/English. I have detailed specific concern in the main manuscript submission document.

**Do you want your identity to be public for this peer review?** For information about this choice, including consent withdrawal, please see our Privacy Policy

Reviewer #1: **Yes: ** Kaleem Tariq, Abdul Wali Khan University Mardan, Pakistan

Reviewer #2: No

---

## [Author Response · Author response to Decision Letter 1]

18 Apr 2025

To the Referees and Editor,

We were pleased to receive your comments for improving our study, and we thank you.

Editor’s comments

Editor: 1. Please ensure that your manuscript meets PLOS ONE's style requirements, including those for file naming.

Authors: We thank the editor for the comments. We checked that our manuscript meets PLOS ONE's style requirements.

Editor: 2. In the online submission form, you indicated that raw data will be available from the authors under request. All PLOS journals now require all data underlying the findings described in their manuscript to be freely available to other researchers, either 1. In a public repository, 2. Within the manuscript itself, or 3. Uploaded as supplementary information.

Authors: We modified the information about raw data and uploaded the dataset to a public repository (Raw data are now available from Figshare: 10.6084/m9.figshare.28777601).

Editor: 3. Please review your reference list to ensure that it is complete and correct. If you have cited papers that have been retracted, please include the rationale for doing so in the manuscript text, or remove these references and replace them with relevant current references. Any changes to the reference list should be mentioned in the rebuttal letter that accompanies your revised manuscript. If you need to cite a retracted article, indicate the article’s retracted status in the References list and also include a citation and full reference for the retraction notice.

Authors: We checked that the reference list was complete and correct. We modified only the Reference 20 following the Editor suggestion (see comments below).

Additional editor comments:

Editor: 1) The quality of the images in the Figures needs to be improved (also pick less closely related colors for the lines Fig2).

Authors: We improved the quality of the images in the Figures and pick less closely related colors for the lines in Figure 2.

Editor: 2) Reference 20 has a different title than the apparent BIORXIV preprint match 'Developing heterospecific Sterile Insect Technique for pest control: insights from the spotted wing fly Drosophila suzukii'. Please explain. What is the status of this manuscript as well as its relationship to this submission? Please confirm that there is no overlap in the data.

Authors: We apologize for the typo, which has been corrected in the revised version of the manuscript. The correct title is 'Developing heterospecific Sterile Insect Technique for pest control: insights from the spotted wing fly Drosophila suzukii'. The incorrect title referred to an earlier version of the preprint, which was subsequently updated following the revisions suggested by the journal to which we submitted that article. We confirm no overlap in the data since the two articles have completely different data sets.

Editor: 3) You will need to fix your data availability statement. Making raw data only available upon request is not compatible with your answer 'Yes' to the data availability question. You can use supplements or a public data repository to fix this or answer 'No' to the question and provide acceptable justification.

Authors: We fixed our data availability statement. Raw data are now available from Figshare: 10.6084/m9.figshare.28777601.

Referee: 1

Reviewer: The study focuses on optimizing the irradiation dose for Drosophila melanogaster males to achieve maximum sterility while minimizing the impact on insect fitness. The authors attempt to correlate these findings with their previous research on the Heterospecific Sterile Insect Technique (hSIT) against Drosophila suzukii. However, no clear relationship between the current study and their previous hSIT findings was established. If an integrated approach combining hSIT and irradiation is required for managing D. suzukii, the experiments should optimize irradiation doses following heterospecific mating. This would allow for a comprehensive evaluation of key parameters, including fertility, sterility, fitness, and courtship behaviors.

Authors: We thank the reviewer for this comment. In our previous studies we investigated reproductive interference between un-irradiated D. melanogaster males and D. suzukii females, obtaining promising results (Cerasti et al., 2023). For this reason, our research has progressed toward evaluating the reproductive interference caused by irradiated D. melanogaster males, bringing us closer to a potential SIT-based application. We included this information in the revised version of the manuscript (lines 110-115 of the tracking version of the manuscript).

Cerasti F, Mastrantonio V, Dallai R, Cristofaro M, Porretta D. Applying Satyrization to Insect Pest Control: The Case of the Spotted Wing Drosophila, Drosophila suzukii Matsumura. Insects. 2023 Jun 19;14(6):569.

Reviewer: If fitness is compromised in D. melanogaster males during heterospecific mating (line 68-72) with D. suzukii females, the males are already weakened before undergoing irradiation. Thus, D. melanogaster males should undergo irradiation after heterospecific mating rather than using untreated males for a more accurate assessment of cumulative fitness trade-offs.

Authors: The sentence in lines 68-72 is a general description of how reproductive interference acts. The fitness cost we refer to does not affect D. melanogaster males, but rather the D. suzukii species, as confirmed by our previous studies, which investigated reproductive interference between the two species without exposing D. melanogaster males to irradiation (Cerasti et al., 2023).

Cerasti F, Mastrantonio V, Dallai R, Cristofaro M, Porretta D. Applying Satyrization to Insect Pest Control: The Case of the Spotted Wing Drosophila, Drosophila suzukii Matsumura. Insects. 2023 Jun 19;14(6):569.

Reviewer: In my view, the current findings cannot be directly correlated with previous hSIT experiments. Below are my main suggestions to enhance the study's impact: 1. Title Revision: The title should be rephrased for clarity. I suggest the following: "Optimizing Irradiation Dose for Drosophila melanogaster Males to Enhance Heterospecific Sterile Insect Technique Against Drosophila suzukii."

Author: We agree with the title suggestion. We modified the title of the manuscript.

Reviewer: 2. Clarify the Study's Importance: If hSIT alone results in complete sterility, why is an optimized irradiation dose necessary? Please address the significance of this study in both the abstract and introduction sections.

Authors: We thank the reviewer for the comment. In this h-SIT the sterility in the target pest population is ensured by the reproductive post-zygotic isolation mechanisms between the released D. melanogaster males and the wild D. suzukii females. Male sterilization avoids potential adverse environmental effects in the natural population dynamics of D. melanogaster. As suggested, we addressed it in the abstract (lines 26-29 of the tracking version of the manuscript) and introduction (lines 115-119 of the tracking version of the manuscript) sections.

Reviewer: 3. Expand on Fitness Costs (Introduction, Line 68): The authors mention that hSIT results in fitness costs for one or both interacting individuals. Please elaborate on how these fitness costs manifest and identify which traits are affected. If flight ability is compromised, it could reduce the feasibility of field releases.

Authors: Thank you for the suggestion. Reproductive interference is a heterospecific interaction that can result in a fitness cost for one or both species involved. In general, the fitness costs are directly tied to behaviors or outcomes related to mating, fertilization, or reproductive success, and flight ability or general physiological performance are not considered direct targets of RI costs. We highlighted this issue in the revised version of the paper and included key references (lines 90-92 of the tracking version of the manuscript).

Reviewer: 4. Quantify Offspring Reduction (Line 87): Please specify the level of offspring reduction achieved in your previous hSIT studies against D. suzukii. If 100% sterility was not achieved through hSIT alone, highlight how this justifies the need for irradiation to enhance hSIT's effectiveness.

Authors: The mating between D. melanogaster males (irradiated and un-irradiated) and D. suzukii females leads to 100% sterility as there is complete post-zygotic isolation. The sentence: “Furthermore, under laboratory conditions, D. melanogaster males irradiated at 60 and 80 Gy were able to court and mate with D. suzukii females, leading to a significant reduction in D. suzukii offspring.” refers to our previous study where we investigated whether reproductive interactions between D. melanogaster males and D. suzukii individuals incur fitness costs for D. suzukii females. We observed a significant effect of D. melanogaster males on the D. suzukii offspring. In fact, the mean number of the individuals originating from five pairs of D. suzukii was 37.95 (± 3.62) (± standard error) in the control tests without the presence of D. melanogaster males, while it was 16.11 (± 5.58) and 13.89 (±3.69), when 40 and 60 irradiated D. melanogaster males were added, respectively (Cerasti et al., 2025). The sentence “However, the induced sterility in D. melanogaster males was not complete at these doses” refers to the sterility experiment where we tested the induced sterility in D. melanogaster males irradiated at 60 and 80 Gy mated with un-irradiated D. melanogaster females. At these two doses, we did not achieve 100% sterility in males (Cerasti et al., 2025). For this reason, in the current study under revision, we tested whether higher irradiation doses could lead to higher male sterility without reducing male performance.

Cerasti F, Cristofaro M, Mastrantonio V, Scifo J, Verna A, Canestrelli D, et al. Developing heterospecific Sterile Insect Technique for pest control: insights from the spotted wing fly Drosophila suzukii [Internet]. bioRxiv; 2025 [cited 2025 Feb 20]. p. 2024.09.05.611447. Available from: https://www.biorxiv.org/content/10.1101/2024.09.05.611447v2

Reviewer: 5. Clarify Behavioral Analysis: Since courtship behavior is complex, and you specifically analyze the time males spend with females, it would be more precise to state: "We analyzed the time spent by male D. melanogaster with D. suzukii females" rather than using the broader term "courtship behavior.

Authors: We thank the Reviewer for this comment. We agree with the reviewer that courtship behavior encompasses complex interactions. In our study, we specifically focused only on the time spent by males courting D. suzukii females. For this reason, we modified the text following your suggestion (lines 130-132; 199; 325 of the tracking version of the manuscript).

Referee 2

Reviewer: The science and experimental design is appropriate for the research questions being addressed. However, there is a need to improve on the presentation of information including methods, results and discussion. The major concern is on the writing style/English. I have detailed specific concern in the main manuscript submission document.

Authors: We thank the Reviewer for the work done on the manuscript. The Reviewer detailed specific concern in the main manuscript submission document, and we included all changes in the revised version of the paper (see tracking version of the manuscript).

---

## [Decision Letter · Decision Letter 1]

Dear Dr. CERASTI,

Thank you for submitting your manuscript to PLOS ONE. After careful consideration, we feel that it has merit but does not fully meet PLOS ONE’s publication criteria as it currently stands. Therefore, we invite you to submit a revised version of the manuscript that addresses the points raised during the review process.

We look forward to receiving your revised manuscript.

Kind regards,

Herman Wijnen, Ph.D.

Academic Editor

PLOS ONE

Journal Requirements:

Additional Editor Comments:

The manuscript is much improved and almost ready to be accepted. I do recommend that you revise your figures to increase the font sizes so that they can be included in the final manuscript without having to take up disproportionate amounts of space. Once this is done, I am happy to accept the manuscript.

Reviewers' comments:

Reviewer's Responses to Questions

**Comments to the Author**

Reviewer #1: All comments have been addressed

2. Is the manuscript technically sound, and do the data support the conclusions?

Reviewer #1: Yes

3. Has the statistical analysis been performed appropriately and rigorously?

Reviewer #1: Yes

4. Have the authors made all data underlying the findings in their manuscript fully available?

Reviewer #1: Yes

5. Is the manuscript presented in an intelligible fashion and written in standard English?

Reviewer #1: Yes

Reviewer #1: The authors have revised the manuscript based on the provided suggestions, and the updated version is suitable for publication in PLOS ONE.

**Do you want your identity to be public for this peer review?** For information about this choice, including consent withdrawal, please see our Privacy Policy

Reviewer #1: **Yes: ** Kaleem Tariq, Abdul Wali Khan University Mardan, Pakistan

---

## [Author Response · Author response to Decision Letter 2]

12 Jun 2025

Response to the reviewer's and Editor's comments

We sincerely thank the Academic Editor and the reviewers for their thorough evaluation and constructive feedback throughout the review process. We are pleased that the revised version of our manuscript was well received.

As requested by the Academic Editor, we have revised the figures by increasing font sizes to enhance readability without affecting the overall layout. All figure files have been updated accordingly. Apart from these changes, we made a single minor correction to a word in the main text for clarity.

We hope that the revised version meets the final requirements for publication in PLOS ONE.

Sincerely,

Flavia Cerasti, Ph.D.

Corresponding Author

Department of Environmental Biology, Sapienza University of Rome

---

## [Editor Report · Decision Letter 2]

Optimizing irradiation dose for Drosophila melanogaster males to enhance heterospecific Sterile Insect Technique (h-SIT) against Drosophila suzukii

PONE-D-25-09324R2

Dear Dr. Cerasti,

We’re pleased to inform you that your manuscript has been judged scientifically suitable for publication and will be formally accepted for publication once it meets all outstanding technical requirements.

Kind regards,

Herman Wijnen, Ph.D.

Academic Editor

PLOS ONE
---

## [Editor Report · Acceptance letter]

PONE-D-25-09324R2

PLOS ONE

Dear Dr. Cerasti,

I'm pleased to inform you that your manuscript has been deemed suitable for publication in PLOS ONE. Congratulations! Your manuscript is now being handed over to our production team.

Kind regards,

on behalf of

Dr. Herman Wijnen

Academic Editor

PLOS ONE